# Plants as Biofactories for Therapeutic Proteins and Antiviral Compounds to Combat COVID-19

**DOI:** 10.3390/life13030617

**Published:** 2023-02-23

**Authors:** Corbin England, Jonathan TrejoMartinez, Paula PerezSanchez, Uddhab Karki, Jianfeng Xu

**Affiliations:** 1Arkansas Biosciences Institute, Arkansas State University, Jonesboro, AR 72401, USA; 2Molecular Biosciences Program, Arkansas State University, Jonesboro, AR 72401, USA; 3Department of Biological Sciences, Arkansas State University, Jonesboro, AR 72401, USA; 4College of Agriculture, Arkansas State University, Jonesboro, AR 72401, USA

**Keywords:** coronavirus, COVID-19, antivirals, vaccines, plant production, molecular farming, recombinant proteins, secondary metabolites

## Abstract

The outbreak of coronavirus disease 2019 (COVID-19) caused by severe acute respiratory syndrome coronavirus 2 (SARS-CoV-2) had a profound impact on the world’s health and economy. Although the end of the pandemic may come in 2023, it is generally believed that the virus will not be completely eradicated. Most likely, the disease will become an endemicity. The rapid development of vaccines of different types (mRNA, subunit protein, inactivated virus, etc.) and some other antiviral drugs (Remdesivir, Olumiant, Paxlovid, etc.) has provided effectiveness in reducing COVID-19’s impact worldwide. However, the circulating SARS-CoV-2 virus has been constantly mutating with the emergence of multiple variants, which makes control of COVID-19 difficult. There is still a pressing need for developing more effective antiviral drugs to fight against the disease. Plants have provided a promising production platform for both bioactive chemical compounds (small molecules) and recombinant therapeutics (big molecules). Plants naturally produce a diverse range of bioactive compounds as secondary metabolites, such as alkaloids, terpenoids/terpenes and polyphenols, which are a rich source of countless antiviral compounds. Plants can also be genetically engineered to produce valuable recombinant therapeutics. This molecular farming in plants has an unprecedented opportunity for developing vaccines, antibodies, and other biologics for pandemic diseases because of its potential advantages, such as low cost, safety, and high production volume. This review summarizes the latest advancements in plant-derived drugs used to combat COVID-19 and discusses the prospects and challenges of the plant-based production platform for antiviral agents.

## 1. Introduction

The outbreak of coronavirus disease 2019 (COVID-19) in Wuhan, China in late December 2019 has posed a serious global public-health emergency [1]. The disease is caused by a highly transmissible and pathogenic coronavirus, named severe acute respiratory syndrome coronavirus-2 (SARS-CoV-2), which causes respiratory disease associated with high fever, difficulty breathing, and pneumonia, etc. [2,3]. As of 14 February 2023, more than 677 million people have been infected by SARS-CoV-2 globally, of which around 6.78 million lives were claimed (Worldometers.info). Equally damaging has been the global economic shutdown for fear of the threat of SARS-CoV-2 transmission. With the recent lifting of the “Zero-COVID Dynamics” policy in China, many more people will be infected, and mortality will continue to increase [4].

SARS-CoV-2 is an enveloped RNA virus with a single-stranded, positive-sense genome of ~29.9 kB in size (Figure 1) [5]. The virus consists of four major structural proteins, named spike (S), nucleocapsid (N), envelope (E), and membrane proteins (M) [1,2]. The S protein which is present as a crown-like spike on the outer surface of the virus plays a major role in viral entry into mammalian cells [6]. Specifically, the virus uses the receptor binding domain (RBD) on the S protein to interact with human angiotensin-converting enzyme 2 (ACE2) receptor as a critical initial step to enter target cells [3,7]. Therefore, both the S protein of the virus (particularly RBD) and the ACE2 of human cells have a potential target to develop therapeutics to prevent SARS-CoV-2 infection [8,9,10,11]. Since the start of the pandemic three years ago, the circulating SARS-CoV-2 virus has been constantly mutating with the emergence of multiple variants (Alpha, Beta, Gamma, Delta, Omicron) [12], which makes the control of the COVID-19 pandemic more difficult. The Omicron variant, after first being identified in South Africa in November 2021, has rapidly spread worldwide, outcompeting other variants, and becoming the predominant one for the time being [13]. The BA.5 subvariant of Omicron, which was the most prevalent coronavirus strain worldwide in 2022, has been found to escape the majority of existing SARS-CoV-2 neutralizing antibodies [12]. Fortunately, infections by Omicron were significantly less severe than those caused by Delta and other previous variants [13]. However, its immune evasiveness and high transmissibility pose a great threat to the global healthcare system [14]. Towards the end of 2022, three Omicron subvariants, BQ.1 and BQ.1.1, and then XBB.1.5 became the dominant strains in the USA, overtaking BA.5 [15]. Therefore, effective prevention and treatments of COVID-19 disease, particularly for people with risk factors for serious illness, are still essential.

Great efforts have been made in the past 2 to 3 years to counteract the spread of the virus through development of vaccines [16], immune-based therapy [17], antiviral therapy [18], and natural remedies [19]. Pharmaceuticals of two general types, including biologics or “big molecules” (e.g., nucleic acids, monoclonal antibodies, therapeutic peptides, etc.) and antiviral chemical compounds or “small molecules” (e.g, Remdesivir, Olumiant, and Paxlovid, etc.) have been developed to prevent and treat COVID-19 (Figure 1). So far, COVID-19 vaccines based on messenger RNA (mRNA) (Pfizer/BioNTech, USA/Germany and Moderna, USA) [20], adenovirus vectors (Johnson & Johnson, USA; AstraZeneca, UK and Sputnik V) [21], subunit protein vaccine (Novavax, USA) [22], and inactivated virus (Sinopharm/Sinovac, China) [23] have been approved or granted Emergency Use Authorization (EUA) by the Food and Drug Administration (FDA) for vaccination in the USA and other countries. These vaccines have been effective at protecting people from getting seriously ill, being hospitalized, and even dying [24]. Almost at the same time, five anti-SARS-CoV-2 monoclonal antibodies (mAbs), including bamlanivimab plus etesevimab (Eli Lilly, USA), casirivimab plus imdevimab (Regeneron, USA), sotrovimab (GlaxoSmithKline, UK and Vir Biotechnology, USA), tocilizumab (Genentech, USA), and bebtelovimab (Eli Lilly, USA) were developed in the USA and have received EUA from the FDA for treating mild-to-moderate COVID-19 [25,26]. Among them, bebtelovimab is the only one that has shown remarkably preserved in vitro activity against all SARS-CoV-2 variants, including the Omicron subvariants BA.4 and BA.5 (but is not effective for BQ.1 and BQ.1.1. subvariants) [27]. In December 2022, tocilizumab became the first monoclonal antibody fully approved by the FDA for COVID-19 treatment. On the other hand, some traditional chemical drugs (small molecules) have been re-evaluated or re-purposed for their potential as an antiviral drug candidate against SARS-CoV-2. Remdesivir (Gilead, USA) [28] and Olumiant (baricitinib) (Eli Lilly, USA) [29] represent the first and second drugs fully approved by the FDA for treatment of hospitalized COVID-19 patients. Early in 2022, the FDA issued an EUA for the emergency use of Paxlovid (Pfizer, USA) and molnupiravir (Emory University, Ridgeback Biotherapeutics, and Merck, USA) for the treatment of mild-to-moderate COVID-19 in adults [30].

Various production platforms, mainly chemical synthesis and mammalian cell culture were utilized to manufacture the aforementioned pharmaceuticals (small and big molecules) to combat COVID-19. Plants have provided a promising alternative production platform for both natural bioactive compounds and recombinant therapeutics [31]. Plants naturally produce a diverse range of bioactive small molecules, such as alkaloids [32], flavonoids [33], terpenoids [34], and phenolic compounds [35], which are the source of countless pharmaceutical compounds for treating various diseases. Many bioactive compounds from medicinal plants, for example, those extracted from *Artemisia annua* L., *Curcuma longa* and *Tripterygium wilfordii* have been demonstrated to exhibit significant activities against SARS-CoV-2 through interfering with every step of the interaction of the virus with its host cells [36,37]. On the other hand, plants can also be genetically engineered to produce heterologous proteins (biologics) for therapeutic applications, termed “molecular farming [38].” Plants bring advantages in safety, scalability, and cost over other eukaryotic systems and have proven effective in mediating the post-translational processing required for many complex proteins [38,39]. Additionally, molecular farming in plants could facilitate rapid production of biologicals at a large scale, as demanded in the case of the COVID-19 pandemic. Extensive research has been performed to produce therapeutics against SARS-CoV-2 in plant systems, including vaccines, antibodies and other immunoadhesins. Notably, in early 2022, Canada-based biotech company Medicago announced that it had gained approval in Canada for its two-dose COVID-19 vaccine Covifenz^®^, an adjuvanted plant-made virus-like particles (VLP) vaccine (www.medicago.com). It represents the first COVID-19 vaccine produced by plant-based protein technology, and the promising results from a Phase III study were recently published [40].

There are many recent reviews published on the use of plant-based agents for the prevention and cure of COVID-19 [41,42,43,44]. However, these articles usually focus on one specific type of plant-produced antiviral agent, particularly bioactive natural compounds (small molecules). There is a lack of an updated review on the antiviral therapeutic proteins (big molecules) produced by plant molecular faming. In addition, most of the review articles are not up to date, because the SARS-CoV-2 virus has been mutating constantly and the therapies for COVID-19 advance rapidly. This review summarizes the latest advancements in plant-derived pharmaceuticals (both big molecules and small molecules) used to fight against SARS-CoV-2 and discusses the prospects and challenges of the plant-based production platform for antiviral agents.

## 2. Plant Produced Biopharmaceuticals (Biologics) against SARS-CoV-2

Plant-based expression systems, or plant molecular farming, have emerged as a promising alternative for the production of biologics. As eukaryotic organisms, plant hosts are able to perform correct post-translational modifications, such as glycosylation, allowing the development of *authentic* biologics with their efficacy being similar to those produced using other expression systems, such as mammalian or yeast-based cell cultures [38,39]. Plant-produced biologics are also regarded as safe because they do not pose the risk of introducing human and animal pathogens into biopharmaceuticals [45]. In addition, plant expression systems, particularly transient expression systems, could prompt rapid (4–8 weeks) manufacturing of target biologics on a large scale [46,47], which meet emergency demands, such as in the case of the COVID-19 pandemic. Given the aforementioned factors, plant-based expression systems have been actively adopted by pharmaceutical manufacturers. A wide range of recombinant proteins, such as vaccines, antibodies, hormones, cytokines, therapeutic enzymes, and nutritional proteins have been produced via stable and transient expression in entire plants or plant cell cultures [45,48]. The first plant-produced biologic for human use, taliglucerase alfa (Elelyso^®^), was approved by the FDA in 2012 for the treatment of Gaucher disease [49]. In 2019, a plant-produced influenza virus vaccine completed Phase III clinical trials with encouraging results [50]. In early 2022, the plant-made COVID-19 vaccine, Covifenz^®^, won first approval in Canada [40]. These successes have revived people’s interest in plant-based production of biologics for human use. To combat COVID-19, plants have been used to produce vaccines [51], monoclonal antibodies [52], and other biologics that block the interactions between ACE2 and the S proteins, such as soluble ACE2 [53] and its fusion with the Fc region of human IgG1 (ACE2-Fc) [54]. In addition, plant-produced antiviral lectin has also been tested for inhibition of SARS-CoV-2 infection [55] (Figure 2).

### 2.1. Plant-Produced Vaccines

Although traditional inactivated viral vaccines and the new adenovirus vector- and mRNA-based vaccines haven been approved and widely used in the world to combat the pandemic, other types of modern vaccines, such as the protein subunit [56] or virus-like particle (VLP) varieties [57], have multiple advantages over currently used vaccines [24]. The minimum requirement for either type of vaccine is the genetic sequence of a single viral antigen rather than the genetic sequence of either virus [24]. This is safer for recipients of the vaccine, because lone antigens cannot cause or spread disease, and safer for scientists researching and manufacturing the vaccine, since no handling of live virus is required once the antigen has been sequenced [58]. SARS-CoV-2 replicates by infecting human cells via the interaction of the RBD on the viral S protein with ACE2 receptors on human cells [3], therefore, the S protein, particularly RBD, has become the focus of vaccine development efforts in the pandemic [24]. A full list of subunit vaccines and VLP vaccines that have reached or passed Phase I human trials according to the COVID-19 vaccine tracker website is available in Appendix A. Plants, either whole plants or cell suspension cultures, are suitable for producing either type of vaccine [42]. A recent comprehensive review of the use of plant-based vaccines for the prevention and cure of human viral diseases can be found in the literature [44,47,59,60]. So far, a few plant-based subunit or VLP vaccines have been developed and some of them have moved to clinical trials (Table 1).

#### 2.1.1. Plant-Produced Subunit Vaccines

In their simplest form, subunit vaccines require only a viral protein capable of eliciting an immune response and an adjuvant. These proteins are capable of eliciting a response from B cells, helper T cells, and cytotoxic T cells, but this response is weak relative to traditional inactivated viral vaccines and necessitates the addition of an adjuvant [58]. A subunit vaccine developed by Novavax (USA) has already been granted EUA by the FDA in 2022 [22]. Studies have showed that this subunit vaccine was about 90% effective in preventing SARS-CoV-2 infections [74](Centers for Disease Control and Prevention, CDC, USA), which is similar to the efficacy of Moderna (94%) and Pfizer (95%) and better than Johnson & Johnson (66%) [75].

Plant expression platforms, mainly transient expression with *Nicotiana benthamiana*, have been used to produce subunit vaccines against SARS-CoV-2 (Table 1). The RBD alone and its fusion with the Fc region of human IgG1 (RBD-Fc) are utilized as an antigen to develop subunit vaccines. The recombinant RBD and RBD-Fc showed specific binding to human ACE2 receptor [51,70,71,72]. In animal tests, the plant-produced RBD and RBD-Fc antigens elicited potent neutralizing responses in mice and non-human primates [51,70]. In order to increase the immunogenicity of the antigen, RBD fused to flagellin of *Salmonella typhimurium* (Flg), known as mucosal adjuvant, was also transiently expressed with *N. benthamiana* using a self-replicating viral vector [76]. As an alternative to the transient expression platform, tobacco BY-2 and *Medicago truncatula* A17 cell suspension cultures were also used to stably express both full-length S protein and RBD [73]. The results showed that recombinant S protein and RBD could be secreted into the culture medium, which facilitated the subsequent purification and reduced the downstream processing costs. This represents the first report on the stable expression of SARS-CoV-2 antigen protein with plant cell culture system, though the bioactivity of the expressed proteins was not assessed.

Plant-produced subunit vaccines have been moved to commercial development. Of particular interest is the subunit vaccines developed by Baiya Phytopharm Co., Ltd. (Bangkok, Thailand), trade names Baiya SARS-CoV-2 Vax 1 and Baiya SARS-CoV-2 Vax 2, that utilizes a *N. benthamiana*-produced RBD as its antigen. A publication on preclinical results states that the RBD protein has been modified by fusing it with the Fc region [51]. When used with alum as adjuvant, Vax 1 induced potent immunological responses in both mice and cynomolgus monkeys [69,77,78]. Vax 1 was also reported to show 100% efficacy against infection in K18-hACE2 mice [79]. The efficacy of, and adjuvant for, the Vax 2 variant has not been revealed so far.

#### 2.1.2. Plant-Produced VLP Vaccines

VLP vaccines make use of one or more viral structural proteins that are capable of self-assembling into nanostructures that mimic the size and shape of a virus [80]. The building blocks of the particle may serve as viral antigens capable of eliciting an immune response and the shape of the overall VLP may conform to a pathogen-associated molecular pattern recognized by the immune system [81]. Where a true virus has a cavity that contains its genetic material, VLPs have a hollow cavity that may be used to deliver small molecules to further enhance the immune response triggered by the vaccine [82].

According to the COVID-19 Vaccine Tracker website (https://covid19.trackvaccines.org/, accessed on 15 January 2023), maintained by scientists at McGill University, at the time of writing only one plant-produced SARS-CoV-2 vaccine has been approved for use. As detailed in Table 1, this VLP vaccine, with the trade name of Covifenz^®^ (Medicago, Canada), has only been approved for use in Canada and it is in Phase III trials in several others. Three other plant-produced vaccines have reached the point of clinical trials, but none of these have yet passed the Phase III trials. Medicago’s VLP vaccine utilizes the recombinant, full-length S protein from an original SARS-CoV-2 strain. These proteins associate into trimers within a lipid membrane from the cell membranes of the host *N. benthamiana* cells. The protein contains modifications made to improve the stability of the protein as well as increase the formation of VLPs [83]. According to Phase III human trials, the vaccine efficacy was 69.5% against symptomatic infection and 78.8% against infection with symptoms ranging from moderate to severe [40]. According to the Canadian government, Covifenz^®^ is administered in two doses 21 days apart and alongside the adjuvant AS03 [61].

Kentucky Bioprocessing’s VLP vaccine, trade name KBP-201, utilizes a recombinant S protein’s RBD and inactivated tobacco mosaic virus. The S protein’s RBD, serving as the antigen, and tobacco mosaic virus, serving as the VLP’s structural component, are each expressed in *N. benthamiana*, and chemically conjugated together following purification [65]. KBP-201’s RBD is fused with the Fc domain of human IgG1 to improve protein stability and an *N. benthamiana* extensin peptide to allow protein secretion and folding in the host species [65]. Preclinical trials with K18-hACE2 mice showed efficacies of 71.4% and 100%, for one and two doses effectively, against lethal infection [64]. The combined Phase I/II trial listed on ClinicalTrials.gov details a two dose regimen 21 days apart using cytosine phosphoguanine as an adjuvant [63].

In addition to Covifenz^®^ and KBP-201, more potential SARS-CoV-2 vaccines (VLP) from iBio, Inc. (Bryan, TX, USA): IBIO-200, IBIO-201 and IBIO-202 have been reported as in pre-clinical trials in separate review publications [44,67]. However, tracking the progress of potential SARS-CoV-2 vaccines that are still in pre-clinical stages poses certain challenges that prevent us from providing accurate, current information on their status. While success in benchtop and animal models may be publicized by academic labs, corporate labs may restrict publications until entry into clinical trials. This is the case with iBio’s VLP vaccines. The citations provided for the status were from iBio’s website (ibioinc.com), a news media website, or another review publication over the same topic rather than a direct, peer reviewed article from scientists responsible for the research. A press release by iBio, dated after the publication of these reviews, has stated that the company will no longer continue development of IBIO-202 and none of the three vaccines appear on the company’s public pipeline [84,85].

### 2.2. Plant-Produced Antibodies

Rather than providing the long-term protection of a vaccine, therapeutic antibodies can be used in the moment to treat people infected by a disease. mAbs targeting epitopes on the virus or infected cells may, alone or as a cocktail, reduce the viral load and thereby reduce the severity of symptoms experienced by the patient [86]. mAb-based therapeutics against the S protein have been shown to be effective treatments for SARS-CoV-2 infection, especially the original viral strain. Up to now, five kinds of FDA approved (EUA) antibodies, either alone or as a mAb cocktail, have been developed to treat COVID-19. However, the current mAbs produced in mammalian cells are expensive and might be unaffordable for many [87]. Plants may provide a low-cost and safety-friendly alternative platform to produce efficacious and affordable antibodies against SARS-CoV-2. As a new production platform, plants have already been demonstrated to have the capability of producing mAbs with quality and characteristics matching those produced in mammalian cells [88]. For example, a plant-made anti-HIV mAb has been found to meet all regulatory specifications for human application in a clinical study [89]. However, to the best of our knowledge, at the time of writing no plant-based antibodies for the treatment of SARS-CoV-2 were in clinical trials. The scope of this portion of the review has been restricted to antibodies expressed within plant cells that have been demonstrated, either in vitro or in vivo, to have a neutralizing effect on at least one variant of a SARS-CoV-2 lineage (Table 2).

The first reported plant-made functional mAbs against SARS-CoV-2 were B38 and H4, which were collected from blood sera of a convalescent patient [52]. These antibodies could block binding between the RBD of the virus and the cellular receptor ACE2. Transient co-expression of heavy- and light-chain sequences of both the antibodies in *N. benthamiana* by using a geminiviral vector resulted in rapid accumulation of correctly assembled mAbs in plant leaves. Both mAbs purified from plant leaves demonstrated specific binding to RBD of SARS-CoV-2 and exhibited efficient virus neutralization activity in vitro [90]. Before this, the same research group tried to express another mAb CR3022 in *N. benthamiana*. However, this plant-produced mAb was found to bind to SARS-CoV-2 but fail to neutralize the virus in vitro [71]. These findings provide proof-of-concept for using plants as an expression system to produce SARS-CoV-2 antibodies.

Plant-made H4 was then examined in greater detail by being expressed in the four human IgG subclasses present in human serum (IgG1–4) [91]. Four constructs, each with the same variable region but different heavy chain regions, were adapted for expression in glyco-engineered *N. benthamiana*. H4-IgG3 demonstrated an up to 50-fold superior neutralization ability compared to the other three IgG against live SARS-CoV-2 virus in vivo. Complete protection from cytotoxic effects of infection (NT_100_) using Vero cells was attained with an H4-IgG3 concentration of 5.91 nM.

Using a cocktail of mAbs that bind to complementary neutralizing epitopes represents a strategy to prevent escape of the SARS-CoV-2 mutant from mAb treatment [87]. To develop mAb cocktail-based therapeutics against SARS-CoV-2 in plants, two neutralizing mAbs, CA1 and CB6 were expressed in *N. benthamiana*. The effectiveness of plant-produced mAbs against the original SARS-CoV-2 virus and a member of the Delta lineage was tested in vitro. Both mAbs retained target epitope recognition and neutralized multiple SARS-CoV-2 variants [87]. The half maximal inhibitory concentration (IC_50_) of CA1 was 9.29 nM for the original strain and 89.87 nM against the Delta strain. The IC_50_ of CB6 was 0.93 nM for the original strain and 0.75 nM for the Delta strain [87]. Both also demonstrated neutralizing potential against a mouse adapted strain of SARS-CoV-2 in vitro. It was also shown that one plant-made mAb has neutralizing synergy with other mAbs developed in hybridomas by the authors. A third neutralizing mAb, 11D7, which was a chimeric human IgG, was then expressed in DeltaXFT *N. benthamiana* to produce a mAb with human-like, highly homogenous *N*-linked glycans [92]. Plant-produced 11D7 was found to maintain recognition against the RBD of original, Delta and Omicron strains and neutralizing activity. Because 11D7 neutralizes SARS-CoV-2 through a mechanism not typical among currently developed mAbs, it may be useful in providing additional synergy to existing mAbs cocktails.

### 2.3. Plant-Produced ACE2-Based Biologics

#### 2.3.1. Plant-Produced ACE2-Immunoadhesins

Although vaccines and antibodies have been developed to effectively combat COVID-19 worldwide, the rapid emergence of SARS-CoV-2 variants with altered RBD can severely affect the efficacy of such immunotherapeutic agents [14]. This problem seems to be especially pronounced with the Omicron variants that resist many of the previously isolated monoclonal antibodies [93]. Immunoadhesins, which are antibody-like molecules, make another class of immunotherapeutic agents that may complement the current therapy issue with vaccines and antibodies [94]. Immunoadhesins consist of an engineered binding domain fused to an Fc region of an antibody [95]. In the case of SARS-CoV-2, the viral cellular receptor ACE2 (extracellular domain) can serve as a binding domain for constructing such immunoadhesins, which can then function as a decoy to block the interaction of the virus with cellular ACE2 receptors [54,96]. Fusing ACE2 with the Fc region offers advantages over the treatment with ACE2 alone. This is because the Fc domain can provide effector functions, allowing the recruitment of some phagocytic immune cells and facilitating the activation of the host antiviral immune response through triggering antibody-dependent cellular cytotoxicity (ADCC) and complement-dependent cytotoxicity (CDC). Furthermore, the Fc domain can prolong the half-life, binding affinity and neutralization efficacy of the binding domain [54,97,98]. So far, more than 13 Fc fusion proteins have been approved by the FDA.

In the past 3 years, many ACE2-based immunoadhesins, including the enhanced ACE2 for binding to S protein of SARS-CoV-2 were developed [8,94,96,99,100,101,102,103,104,105]. These ACE2-immunoadhesins were effective in neutralizing multiple SARS-CoV-2 variants, including the Delta and the Omicron variants, suggesting that immunoadhesins-based immunotherapy is less prone to escape by the virus [94]. Again, plants can provide an economic platform to rapidly produce these biologics.

With transient expression in *N. benthamiana*, ACE2-Fc was produced at up to 100 µg/g fresh leaf. The recombinant ACE2-Fc exhibited potent anti-SARS-CoV-2 activity in vitro, and dramatically inhibited SARS-CoV-2 infectivity in Vero cells with an IC_50_ value of 0.84 µg/mL. Furthermore, treating Vero cells with ACE2-Fc at the pre-entry stage suppressed SARS-CoV-2 infection with an IC_50_ (half maximal inhibitory concentration) of 94.66 ug/mL [98].

Because ACE2 is heavily glycosylated and its glycans impact on binding to the S protein and virus infectivity, the ACE2-Fc was also expressed in glycol-engineered *N. benthamiana*. It was found that the recombinant dimeric ACE2-Fc was glycosylated with mainly complex human-type *N*-glycans and showed function in peptidase activity, binding to the RBD of the virus and neutralizing the wild-type SARS-CoV-2 virus [106].

#### 2.3.2. Plant-Produced ACE2 and ACE2-Based Chewing Gum

Besides the ACE2-based immunoadhesins, ACE2 alone could also be developed as a therapeutic to inhibit the virus spread, though there are limitations, such as short circulating half-life [54]. Human soluble (truncated) ACE2 was reported to express in *N. benthamiana* with a high-level yield (about ~750 µg/g fresh leaf). Plant-produced ACE2 could bind to the SARS-CoV-2 S protein. Both glycosylated and deglycosylated forms of ACE2 demonstrated strong anti-SARS-CoV-2 activities in vitro, with an IC_50_ being ~1.0 and 8.48 μg/mL, respectively [53].

Of special interest is the ACE2-based chewing gum developed by Dr. Henry Daniell and his colleagues at the University of Pennsylvania [107,108,109]. This virus-trapping gum contains plant-made CTB-ACE2, which is ACE2 fused with non-toxic cholera toxin subunit B (CTB). CTB-ACE2 is made in chloroplasts of transgenic lettuce. The lettuce was then powdered and blended with cinnamon-flavored chewing gum. The CTB-ACE2 can efficiently bind to both GM1 and ACE2 receptors, effectively blocking binding of the S protein and viral entry into human cells. As oral epithelial cells are enriched with both receptors, this gum was designed to trap and neutralize SARS-CoV-2 in the saliva and diminish the amount of virus left in the mouth. The Phase I/II clinical trial of the chewing gum started in June 2022 (ClinicalTrials.gov Identifier: NCT05433181). If the gum proves safe and effective, it could be given to patients whose infection status is unknown or even for dental check-ups to reduce the likelihood of passing the virus to caregivers [109].

### 2.4. Plant Produced Antiviral Lectins

Lectins from plants and algae, which are carbohydrate-binding proteins of non-immune origin, were earlier found to inhibit several viral diseases, such as HIV, hepatitis C, influenza A/B, herpes, Japanese encephalitis, Ebola, and SARS coronavirus that occurred in 2003 [110,111,112,113]. Recently, some lectins have shown significant activity against SARS-CoV-2 [114,115,116]. For example, Griffithsin, a red algae-derived lectin of 121 amino acids, is a high mannose-specific lectin that has been recognized as a potential viral entry inhibitor [117]. Griffithsin was tested for SARS-CoV-2 entry and found that it could significantly inhibit the SARS-CoV-2 infection in a dose-dependent manner. Remarkably, the IC_50_ of griffithsin was 63 nmol/L, which is about 11-fold more potent than Remdesivir [55]. Other research demonstrated that griffithsin could block the entry of SARS-CoV-2 and its variants, Delta and Omicron, into the Vero E6 cell lines and IFNAR^–/–^ mouse models by targeting the S proteins of the virus [118]. Similarly, recent molecular docking studies have shown that a banana-derived mannose-specific lectin could also neutralize SARS-CoV-2 infectivity [119]. Lectins are natural proteins which are cheap and easily accessible. They have been proven to be active against SARS-CoV-2. However, their clinical application is still hampered by several obstacles. These include the high-cost purification, short stability in the body, potential cytotoxicity and mitogenicity, and the possibility for eliciting deleterious responses in the immune system [120]. Future investigations are needed to develop plant lectins as a new antiviral agent against COVID-19.

### 2.5. Challenges in Commercialization of Plant-Produced Biologics against SARS-CoV-2

Numerous anti-SARS-CoV-2 biologics, including vaccines, antibodies, and other biologics against the virus have been expressed in plant systems, as mentioned above. However, compared with other production systems, such as bacterial and mammalian cell culture, plant systems suffer from a major disadvantage: low production levels of the desired proteins [48]. Additionally, isolation and purification of the recombinant proteins from plant tissues is quite expensive [38]. Although plant systems have proven effective in performing glycosylation required for complex proteins [48,121], there is a major difference in the plant and mammalian glycan structure. The N-linked glycans produced by plants carry two plant specific residues, β-1,2-xylose and core α-1,3-fucose, which are absent from mammalian cell produced proteins [122]. The immunogenicity and allergenicity of plant-specific N-glycans has been a key concern in human therapy [122]. So far, there is only one plant cell produced biopharmaceutical, taliglucerase alfa (Elelyso^®^), approved by FDA. Concerted research efforts based on molecular biology strategies, such as enhancing gene transcription and translation, minimizing post-translational degradation, and glycoengineering to humanize glycosylation, and engineering strategies, such as improving bioreactor design and operation and optimizing the protein purification procedure, are still needed for the commercial success of plant-based production platforms.

## 3. Medicinal Plant-Produced Metabolites (Small Molecules) against SARS-CoV-2

Although some vaccines and mAbs have been successfully developed in the past 2 to 3 years to combat COVID-19 disease, the lack of effective therapeutics against the virus has prompted the shift of some interests toward plant-based therapy. This is because many drugs in use are either plant materials or derived from their bioactive compounds. There is a remarkable prospect of discovering anti-COVID-19 from medicinal plants [123].

Plant-produced secondary metabolites (PSMs) are a rich source of bioactive compounds with a broad spectrum of antiviral activities [37]. Due to their high bioavailability, relatively low cost, and potential for large-scale production, PSMs represent a promising field of study to find new treatments against SARS-CoV-2 [124,125,126]. The potential of PSMs to treat COVID-19 is immense. They can be utilized as prophylactics, antivirals, and even adjuvants to reduce morbidity during COVID-19 treatment [127]. Natural medicines derived from PSMs are usually non-toxic, well-tolerated with minimum side effects, and highly absorptive by the human body [128]. A list of 162 PSMs found in medicinal plants that showed antiviral activity has been published earlier [37]. Among them, around 76 PSMs from different plant species are effective against COVID-19 [37]. These PSMs can be generally classified as polyphenols, alkaloids, flavonoids, coumarins and essential oils, which are able to inhibit main targets in the virus life cycle, including the viral proteins, the lipid envelope and viral nucleic acids [129]. In addition, advanced bioinformatics applications have opened a new arena in predicting PSMs as a potential COVID-19 suppressor [128,130]. In silico analysis has revealed that PSMs could be one of the most valuable drug targets against SARS-CoV-2 [37]. There are many recent review papers published on the PSMs against SARS-CoV-2 [36,37,125,129,131,132,133,134,135,136]. A comprehensive list of medicinal plants and their active compounds with inhibitory activity against SARS-CoV-2 can be seen in the recent reviews [37,131,132,137]. In this section, we will discuss the major antiviral mechanisms of PSMs against SARS-CoV-2 and summarize some of the newly published data involving the application of PSMs in the prevention and treatment of COVID-19 infections.

### 3.1. Antiviral Mechanisms of PSMs

Many PSMs have broad-spectrum antiviral activity. They can inhibit multiple steps in viral infection and replication and have been previously used in the treatment of SARS, MERS, influenza, and dengue virus [131,138]. Specifically, PSMs may function in inhibiting viral proteins, intercalating viral nucleic acids, blocking the ACE2 receptor, and modulating the immune system (Figure 3) [129,132,135,138,139].

#### 3.1.1. Inhibition of Viral Proteins

The major drug targets that have been identified for SARS-CoV-2 through host-virus interaction studies include the SARS-CoV-2 main protease (Mpro), chymotrypsin-like protease (3CLpro), papain-like protease (PLpro), RNA-dependent RNA polymerase (RdRp), helicase Nsp13 and S proteins [36]. These viral proteins are critically involved in the viral replication and transcription process, and thus considered as the most promising targets for drug discovery against SARS-CoV-2 [135]. Since the outbreak of the COVID-19 pandemic, research has been conducted to screen for potential PSMs inhibiting the SARS-CoV-2 proteases, RdRp and other viral proteins using molecular docking analysis. Possible PSM inhibitors against major proteases and helicases of COVID-19 were summarized in recent reviews [129,140]. For example, six FDA-approved antiviral compounds, including Withaferin A, Nelfinavir, Rhein, Withanolide D, Enoxacin, and Aloe-emodin were discovered as possible COVID-19 main protease inhibitors [141]. In other research, the binding of a library of polyphenols with SARS-CoV-2 RdRp was assessed, and the study revealed that epigallocatechin gallate and three theaflavin derivatives could strongly bind to the active site of RdRp with high binding stability and with low toxicity, thus representing an effective therapy for COVID-19 [142]. In addition, in an in vivo study using hamsters as a disease model, extracts of *Perilla frutescens* and *Mentha haplocalyx* were found to be effective in inhibiting viral 3CL protease and RdRp [143], thus these compounds could be further developed as plant-derived anti–SARS-CoV-2 agents.

#### 3.1.2. Intercalation of Nucleic Acids

Some types of PSMs, particularly alkaloids, can directly interact with DNA or RNA and hence stabilize those in single-stranded form. These DNA- or RNA-alkaloids conjugate readily, inhibit their further replication, and consequently prevent viral replication [128]. The well-known alkaloids that have been found to intercalate viral nucleic acids include berberine, emetine, sanguinarine, isoquinoline, beta-carboline, quinoline, paraquinine, dictamine, skimmianine [129]. Due to this mechanism, plant-derived alkaloids represent an important group of PSMs used to combat SARS-CoV-2.

#### 3.1.3. Blocking of ACE2 Receptor

Because SARS-CoV-2 enters human cells through the ACE2 receptor, the simplest way to prevent viral infection is through blocking this receptor. Several PSMs, such as flavonoids, xanthones, proanthocyanidins, and secoiridoids have shown their binding affinities towards ACE2, thus becoming potential natural drugs against COVID-19 [144]. For example, quercetin was found to efficiently inhibit ACE2 [145]; essential oils isolated from lemon and Geranium could significantly downregulate the expression of the ACE2 receptor in human epithelial cells [146].

#### 3.1.4. Immune Modulation

A growing body of clinical data has indicated that COVID-19 may cause a “cytokine storm” in patients due to an extreme inflammatory response, which is also a crucial cause of death from COVID-19. Certain PSMs have shown positive immunomodulation effects against this “cytokine storm”. For example, alcoholic extract of hop (*Humulus lupulus*) and bark of cinnamon (*Cinnamum verum*) was found to inhibit NF-κB (nuclear factor kappa-light-chain-enhancer of activated B cell) which acts as a pro-inflammatory element [147].

### 3.2. Major Classes of PSMs against SARS-CoV-2

PSMs from three major classes: alkaloids, polyphenols and terpenoids/terpenes, have shown activities in preventing and treating COVID-19 infections [123].

#### 3.2.1. Antiviral Alkaloids

Alkaloids represent a large class of PSMs that contain at least one nitrogen atom. According to their biosynthetic pathway alkaloids can be classified into several groups: tropanes, quinolines, indoles, purines, isoquinolines, imidazoles, pyrrolidines, pyrrolizidines and pyridines. The pharmacological effects of these alkaloid compounds include antioxidant, antifungal, antimalarial, antibacterial, and antiviral activities. The antiviral activity of some alkaloids, such as emetine, Ipecac, Macetaxime, tylophorine, and 7-methoxy cryptopleurine, is shown by inhibiting viral proteases, RNA synthesis and protein synthesis [37,124]. In silico screening analysis indicated that some alkaloid compounds, for example 10-hydroxyusambarensine and cryptoquindoline isolated from African medicinal plants, exert anti-SARS-CoV-2 activity through inhibition of 3CLpro [148]. In addition, some alkaloids, such as tetrandrine, fangchinoline, cepharanthine, and lycorine, inhibit the virus through intercalating into nucleic acid and inhibiting spike and nucleocapsid proteins [149]. So far, the alkaloid compounds showing the greatest inspiring antiviral effects against SARS-CoV-2 are papaverine, caffeine, berberine, colchicine, cryptospirolepine, deoxynortryptoquivaline, cryptomisrine, 10-hydroxyusambarensine, emetine, ergotamine, camptothecin, lycorine, nigellone, norboldine, and quinine [133,150]. These compounds could be further developed by being used alone or in combination with other drugs for treating COVID-19.

#### 3.2.2. Antiviral Polyphenols

Polyphenols contain multiple aromatic rings and one or more hydroxyl groups. Polyphenols are broadly classified as flavonoids, lignans, stilbenes and phenolic acids according to the number of aromatic rings they contain and of the structural elements binding these rings together [151]. Numerous polyphenols are considered as antiviral agents. A comprehensive review on the phenolic compounds against SARS-CoV-2 was recently published [152]. Polyphenols show antiviral activities using diverse mechanisms, including intercalating into nucleic acid and inhibiting the activity of protease, helicase and RdRp [152,153]. This is because the hydroxyl group of polyphenols can interact with the positively charged amino groups of proteins and consequently destroy the three-dimensional structure of the protein [123]. In silico analysis revealed that polyphenols can inhibit the Mpro protease and RdRp of SARS-CoV-2 effectively [154]. Flavonoids consisting of two aromatic rings bound together by three carbon atoms comprise the most studied group of polyphenols. Flavonoid compounds were found to be able to inactivate the Mpro protease of SARS-CoV-2 [155]. In addition, the flavonoid scutellarein from the root of *Lamiacaea* was shown to inhibit the NSP13 helicase of SARS-CoV-2 by altering its ATPase activity [156]. In the past three years, many clinical trials of polyphenols as a possible treatment for patients with COVID-19 have been reported [152]. Polyphenols from different plant species have been shown to improve symptoms (fever, chills, cough, myalgia, and tachypnea), increase lymphocyte count, and decrease inflammation, etc. [152].

#### 3.2.3. Antiviral Terpenoids/Terpenes

Terpenoids constitute a large group of PSMs with a broad spectrum of structures and effects. They are lipophilic compounds found in essential oils of many plants. Terpenoids could be used as antioxidant, anti-cancer, anti-inflammatory, antibacterial and antiviral reagents [123]. In terms of antiviral activity, lipophilic terpenoids can disturb the lipid envelope of viruses. Certain terpenes, such as celandine-B, betulinic acid, and ursolic acid have demonstrated strong antiviral effects as they can destroy the lipid layer of the virus [157]. Recent in silico screening indicated that some terpenes from African plants, such as 6-oxoisoiguesterin, 22-hydroxyhopan-3-one and 20-epi-isoiguesterino, could interact with the 3CLpro of SARS-CoV-2, and had binding affinities surpassing that of two reference compounds, lopinavir and ritonavir [148]. Recently, Glycyrrhizin, a triterpenoid saponin from licorice (*Glycyrrhyza glabra*) roots, was reported to be valuable in the treatment of COVID-19 due to its multi-target mode of action, such as binding to ACE2, downregulating proinflammatory cytokines, and stimulating endogenous interferon [158]. In addition, cannabidiol (CBD) from *Cannabis sativa* has been shown to downregulate ACE2 expression in COVID-19 target tissues, thus reducing COVID-19 severity [156].

### 3.3. Potential Anti-SARS-CoV-2 Compounds

Although numerous studies have focused on the inhibition of SARS-CoV and MERS-CoV with PSM compounds in the past years, there are few studies on the direct treatment of COVID-19 disease, which are limited to in silico studies [126]. A compilation of PSMs with potential inhibitory and regulatory activity against SARS-CoV-2 is listed in Table 3. Their chemical structure is shown in Figure 4. These metabolites were curated based on their potential ability to stop, prevent, and treat COVID-19 infections. Only those with potential direct activity against SARS-CoV-2 were considered. Additionally, most PSMs in Table 3 show little toxicity, as indicated by in vitro or in silico evaluations.

Given the novelty of the disease and the shift in attention to other areas, PSMs have not been studied thoroughly for treating COVID-19 and most research is still in early phases, as evidenced by the predominance of in vitro models throughout the table. Therefore, the metabolites presented are only prospects selected by their confirmed in vitro activity and/or their theorized capabilities by in silico models. More research is needed to fully corroborate their action against SARS-CoV-2 in humans. The PSM compounds introduced below are a selection of those with the most advanced stages in research.

#### 3.3.1. Artemisinin

Artemisinin derivatives from *Artemisia annua* L. are effective in treating malaria. They are also well documented as antiviral drugs [176]. The study by Nair et al. (2021) revealed a potent inhibitory action of *A. annua* L. leaf extracts on viral infection in Vero E6 cells with relatively low IC_50_ values [159]. The study also demonstrated that artemisinin is not the main or only metabolite with antiviral properties in the extracts. Furthermore, the metabolites present in the leaf extracts were found effective against five variants of the virus (Alpha, Beta, Gamma, Delta and Kappa) and seemed to have great stability [160]. Dry leaf samples still showed antiviral activity after staying frozen for twelve years [159]. In another study by Zhou et al. (2021), it was found that *A. annua* extracts as well as individual compounds (artemisinin, artesunate, and artemether) all showed inhibitory effects on viral infection of Vero E6 cells, human hepatoma Huh7.5 cells and human lung cancer A549-hACE2 cells. Among them, artesunate proved most potent in different cell types, and it targeted SARS-CoV-2 at the post-entry level [161].

#### 3.3.2. Hesperidin/Hesperetin

Hesperidin and its aglycone hesperetin, compounds isolated from citrus plants, are also regarded as potential antiviral drugs. Both compounds were initially studied due to their predicted interaction in molecular models with ACE2 and TMPRSS2 [165]. These molecules are crucial in the SARS-CoV-2 cell hijacking mechanism. Cheng et al. (2022) applied enzymatic activity assays and in vitro studies with Vero E6 cells and particles of pseudo-virus, demonstrating a strong disruption of ACE2-S protein interaction, but a slight inhibition of TMPRSS2. Nonetheless, their research also revealed another action mechanism for hesperidin and hesperetin—the downregulation of both ACE2 and TMPRSS2 in lung epithelial lung cancer cell lines. These metabolites’ disruptive activity and little cytotoxicity make them prospective agents to prevent further cell entry in infected individuals, allowing for a quicker recovery.

#### 3.3.3. Emetine

Emetine, an alkaloid extracted from the *ipecacuanha* plant (Brazilian root), is another compound with strong antiviral activity [177]. Its effect against other coronaviruses is well documented and SARS-CoV-2 is no exception. Several papers have documented the multifaceted approach of emetine against SARS-CoV-2, inhibiting not only viral entry, but also replication and proliferation [164]. In a study published by Wang et al. (2020) emetine swiftly reduced viral activity in SARS-CoV-2 infected Vero cells and blocked viral entry in pretreated cells [178]. The metabolite disrupts the interaction between viral mRNA and key molecules involved with translation, including ribosomes, viral polymerase RdRp and replication–transcription complex translation initiation factor eIF4E [179,180], as suggested by in silico simulations. In addition, the action of emetine remains effective even at low concentrations, making it a potent drug. However, the researchers also noted that emetine has been associated with cardiac complications and possible cytotoxicity, leaving its viability as a drug yet to be assessed.

#### 3.3.4. Luteolin and Quercetin

Similar to emetine, luteolin and quercetin, which are ancestors of flavonoid natural compounds, exhibit in vitro inhibition against RdRp of SARS-CoV-2 [168]. Previous in vitro studies have also suggested their inhibitory activity against viral protease 3CLpro [71]. Their ability to reduce SARS-CoV-2 replication by preventing translation and post-translational processing makes both metabolites promising treatment drugs for COVID-19.

#### 3.3.5. Panduratin A

Panduratin A, a diarylheptanoid found in *Renealmia nicolaioides* and *Boesenbergia rotunda* can not only prevent infection but also slow viral replication. As discovered by Kanjanasirirat et al. (2020), panduratin A significantly stops the activity of SARS-CoV-2 in Vero E6 and Calu-3 cell lines in pre- and post- infection stages with a performance comparable to the already approved COVID-19 treatment drugs. Additionally, it is important to highlight the minimal toxicity shown by panduratin A, as evidenced by cytotoxicity assays in five different cell lines [171].

#### 3.3.6. Tannins

The case of persimmon-derived tannins has vital importance as it represents one of the few studies to include in vivo models. After the hamster model was treated with persimmon-derived tannins and then inoculated with SARS-CoV-2 viral particles, it remained healthy and presented low antigen and viral load levels in their lungs, as indicated by immunohistochemistry assays and qPCR [172]. By contrast, the control group presented severe lung inflammation and greater pathophysiology and a higher antigen count. Even though the researchers demonstrated the prophylactic potential of persimmon-derived tannins to prevent/regulate SARS-CoV-2 infection, further studies are needed to assess the efficacy of the compound in already infected organisms.

### 3.4. Challenges in Clinical Applications of PSMs against SARS-CoV-2

PSMs from various medicinal plants serve as a treasure of bioactive compounds that have shown promising results against SARS-CoV-2. However, due to the novelty of COVID-19, and lack of experimental evidence and safety studies, the use of most PSMs is still limited [126]. So far, none of the isolated PSM compounds from medicinal plants have been successfully used for clinically treating COVID-19. First of all, there is still a lack of sufficient in vivo and clinical studies to demonstrate the effectiveness of PSM compounds in preventing the viral infection or alleviating symptoms associated with virus infection [181]. In fact, numerous compounds showing high antiviral activity in vitro could be found to be inactive in pre-clinical or clinical trials [138]. Second, because the PSMs are small molecules, they are usually stable and can be delivered orally as plant extracts without the need to purify from other by-products. However, the treatment with plant extracts becomes complicated due to the existence of various compounds. The antiviral ability of individual compounds may be different from their functions in extracts, and they can be additive or synergistic, or even antagonistic [156]. In addition, some PSMs could be toxic at certain levels, so it is necessary to conduct in vitro and in vivo research to evaluate the safety and therapeutic levels of each compound before conducting human clinical trials [131]. Therefore, tremendous research is still needed to find the most effective PSM compounds or a combination of them that would be effective in treating COVID-19 infection.

## 4. Prospects

The COVID pandemic has been affecting the world for three years. It is generally accepted that SARS-CoV-2 will not be fully eradicated. Most likely, the virus will coexist with humans, and the disease will become an endemicity. There is a critical need to develop new and effective pharmaceuticals for prophylactic and therapeutic purposes. The rapid development of vaccines against SARS-CoV-2 has provided a major step forward in reducing COVID-19’s impact, thus representing a scientific victory. However, many people, even those fully vaccinated lose protection over time. This problem seems to be especially pronounced with the Omicron-related variants. Therefore, there is still a pressing need for globally available vaccines that can provide more lasting immunity against current and future coronavirus variants [94]. A ferritin-based COVID-19 nanoparticle vaccine that elicits robust, durable, broad-spectrum neutralizing antisera against known variants of concern, including Omicron BQ.1 and the previous virus version SARS-CoV-1, was just reported [182]. This offers great potential for the rapid response of the emerging SARS-CoV-2 variants and provides versatility for the future development of vaccines against other emerging coronaviruses. In addition, all currently approved vaccines are administered parenterally, which may not be effective in preventing mucosal infection of respiratory pathogens like SARS-CoV-2. Therefore, mucosal COVID-19 vaccines, such as those administered orally or intranasally, would potentially be more effective in offering protection against SARS-CoV-2 infection, because they offer the dual benefit of inducing potent mucosal and systemic immunity [183]. Around 100 mucosal COVID-19 vaccines are in development globally. Among them, 20 have reached clinical trials in humans [184]. In late 2022, two mucosal COVID-19 vaccines that are delivered through the nose or mouth have been approved for use in China and India [185]. These approvals validate the need for mucosal vaccines, though their effectiveness in preventing COVID-19 infection needs be further assessed. In addition to vaccines, other antiviral agents with the potential to limit virus transmission or block infection, including both big molecules and small molecules, will continue to be explored.

Plants provide attractive bioproduction platforms for both recombinant therapeutics and natural bioactive metabolites to combat the COVID pandemic. Molecular farming in plants is an unprecedented opportunity for developing vaccines, antibodies, and other biologics for pandemic diseases because of its potential advantages, such as low cost, safety, and high production volume. As mentioned above, numerous anti-SARS-CoV-2 biologics have been expressed in plant systems. Because many plants or plant products are edible, advances in this area can lead to oral vaccines that can induce mucosal immunity, and that are low cost, easy-to-administer, and have high thermostability [186]. This could be especially applicable for vaccinating people in developing countries, such as those in Africa, where high costs and logistical problems can constrain massive vaccine programs [60]. However, compared to the major expression hosts (bacteria, yeast and mammalian cells), plants are still largely underutilized, mainly due to low productivity and non-human glycosylation [187,188]. Modern molecular biology tools, such as RNAi and the latest genome editing technology, could be exploited to modulate the genome of plant cells to create new plant lines exhibiting improved “traits” for therapeutic protein production [189].

Alternatively, medicinal plants provide a significant prospect for discovering new and effective anti-COVID-19 drugs. The PSMs from medicinal plants have been demonstrated to be powerful in fighting against SARS-CoV-2 as they can interfere with the viral life cycle, including viral entrance, replication, assembly, and virus-specific host targets [190]. Many potential antiviral PSMs against SARS-CoV-2 have been tested in vitro, in vivo or predicted by in silico analysis. However, none of the PSM compounds have been approved by the FDA for treating COVID-19 so far. More pre-clinical and clinical evaluation of the therapeutic effectiveness of these PSMs is a major concern for further development of safe and effective treatments. Because of the novelty of the virus and the disease caused, safety is still a main concern for the use of PSMs [138]. Although compared to synthetic medicines, PSMs are regarded as less toxic because of their natural origin and long-term use as traditional medicines, these compounds may have potential adverse or toxic effects at certain concentrations [36]. Therefore, further investigations, particularly pre-clinical evaluation, are necessary for determining the safe therapeutic dose of each compound before clinical application. In addition, plant metabolomics is currently used as a tool to discover novel drugs from plant resources [37]. Characterization of genes and enzymes involved in secondary metabolic pathways is also very crucial for understanding the biosynthesis of bioactive compounds [191]. This will pave the way for further genetic modifications of medicinal plants to synthesize novel PSM compounds that are most effective in treating COVID-19. Finally, in order to improve the use of PSM compounds, combination treatments, for example the treatments in combination with the FDA-approved anti-SARS-CoV-2 drugs or with the assistance of nanotechnology, may be a promising strategy to develop as they exhibit better synergistic and/or additive effects against COVID-19 [137].

## 5. Conclusions

The COVID-19 pandemic not only caused a public health crisis, but also severely affected the global economy. Although the epidemic has been alleviated to a great extent around the world, the virus will continue to coexist with human beings and constantly mutate. Plants provide a promising bioproduction platform for both recombinant therapeutics (big molecules) and natural bioactive compounds (small molecules) that can be used to combat the virus. “Molecular farming” in plants proposes a superior bioproduction platform for recombinant therapeutics as compared to other eukaryotic systems in terms of safety, scalability, and cost. In the future, advances in this area could also lead to oral vaccines that may be convenient and easy to deploy. Alternatively, plants represent a dramatically underutilized source of bioactive compounds with a broad spectrum of antiviral activities. In vitro, in vivo, and in silico analyses have revealed numerous plant-derived compounds with promising anti-SARS-CoV-2 activity. Therefore, these molecules will be able to develop new natural solutions for treating COVID-19. In summary, it will take the combined efforts of plant genetic engineering and natural plant medicine research to ultimately extinguish this pandemic.

## Figures and Tables

**Figure 1 life-13-00617-f001:**
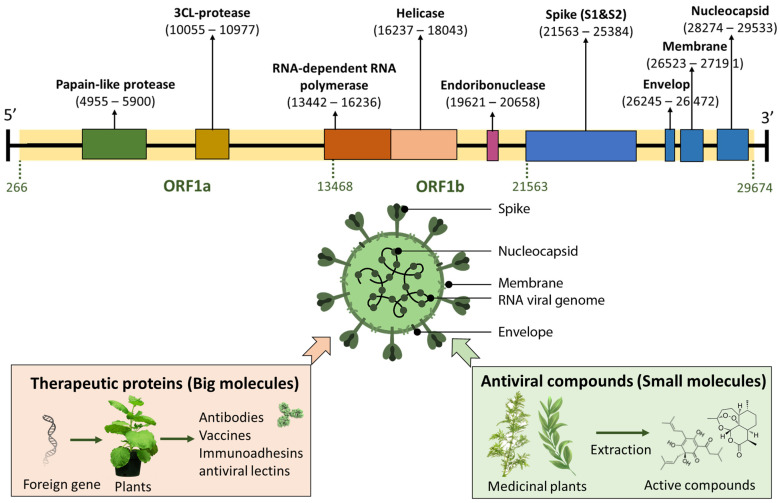
Schematic illustration of the single stranded RNA genome of SARS-CoV-2 (~30 kb) and pharmaceuticals (big molecules and small molecules) developed to prevent and treat COVID-19.

**Figure 2 life-13-00617-f002:**
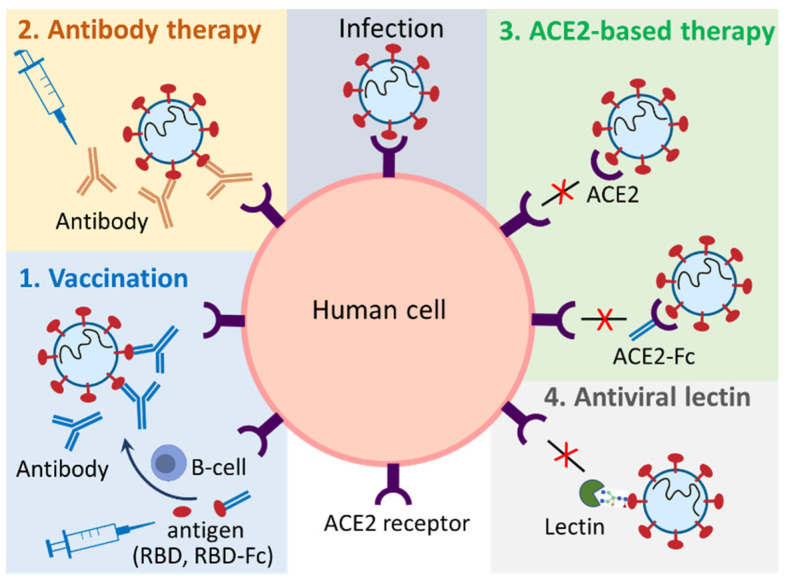
Schematic diagram of the plant-produced biologics functioning in preventing and treating SARS-CoV-2 infection. Plant-produced vaccines (1), antibodies (2), ACE2-based biologics (ACE2-immunoadhesin, ACE2-chewing gum) (3), and antiviral lectins (4) can be used to combat COVID-19.

**Figure 3 life-13-00617-f003:**
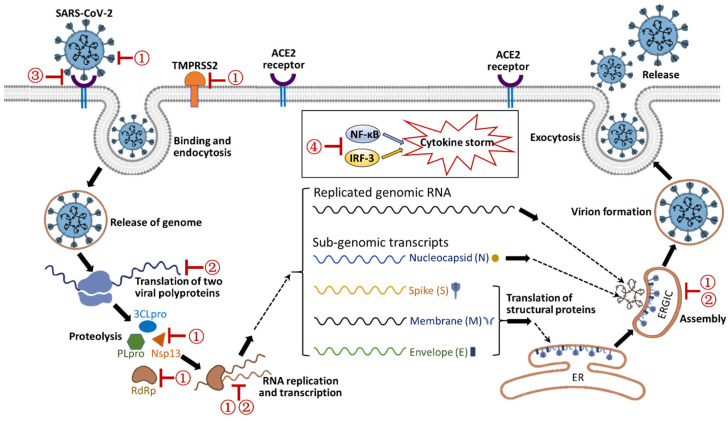
Schematic illustration of anti-SARS-CoV-2 mechanisms of PSMs from medicinal plants. Possible inhibition targets of PSMs during the viral life cycle are indicated by ①: inhibition of viral proteins; ②: intercalation of nucleic acids; ③: blocking of ACE2 receptor; ④: Immune modulation. TMPRSS2: Transmembrane protease, serine 2; PLpro: papain-like protease; 3CLpro: chymotrypsin-like protease; Nsp13: Nsp13 helicase; RdRp: RNA-dependent RNA polymerase; NF-κB: nuclear factor kappa-light-chain-enhancer of activated B cell; IRF3: interferon regulatory factor 3; ER: endoplasmic reticulum; ERGIC: ER-Golgi intermediate compartment.

**Figure 4 life-13-00617-f004:**
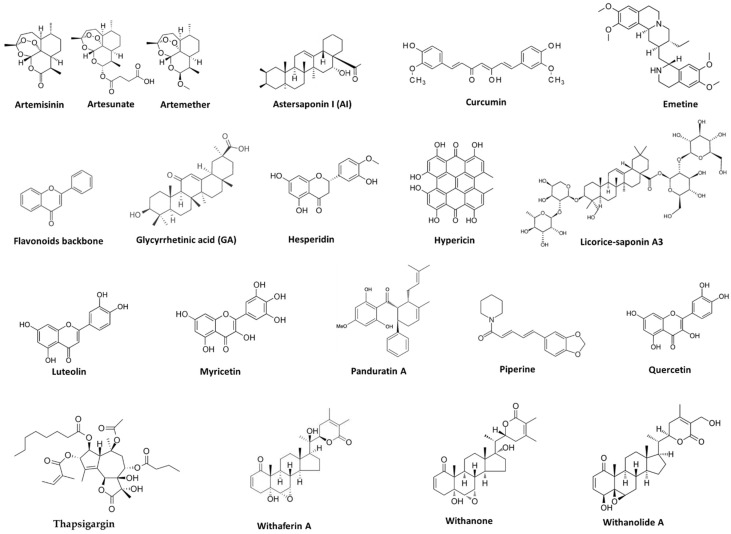
Chemical structures of some bioactive PSM compounds against SARS-CoV-2. Those with defined structures are shown here.

**Table 1 life-13-00617-t001:** Plant-produced SARS-CoV-2 vaccines. Four of them have progressed to clinical trials.

Table	Trade Name	Antigen	Plant	Manufacturer	Efficacy	Commercialization Progress	Source
Virus-like particles	Covifenz	S protein	*N. benthamiana*	Medicago	69.5% to 78.8% (Phase III)	Approved: Canada Phase III Trials: Argentina, Brazil, United Kingdom, USA	[40,61,62]
	KBP-201	RBD	*N. benthamiana*	Kentucky Bioprocessing	100% (K18-hACE2 mice)	Phase I/II Trials: USA	[62,63,64,65]
	IBIO-200, IBIO-201, and IBIO-202	S protein	*N. benthamiana*	iBio, Inc.	n.d.	Pre-clinical trials	[44,66,67]
	n/a	S protein	*N. benthamiana*	n/a	n.d.	no	[68]
Subunit	Baiya SARS-CoV-2 Vax 1	RBD-Fc	*N. benthamiana*	Baiya Phytopharm	100% (K18-hACE2 mice)	Phase I Trials: Thailand	[62,69]
	Baiya SARS-CoV-2 Vax 2	RBD-Fc	*N. benthamiana*	Baiya Phytopharm	Unknown	Phase I Trials: Thailand	[62,69]
	n/a	RBD-Fc	*N. benthamiana*	n/a	n.d.	no	[51]
	n/a	RBD	*N. benthamiana*	n/a	n.d.	no	[70,71,72]
	n/a	S protein, RBD	Tobacco BY-2 and Medicago truncatula A17 cell	n/a	n.d.	no	[73]

n/a: not applicable; n.d.: no data.

**Table 2 life-13-00617-t002:** Plant-produced antibodies against the SARS-CoV-2 virus. Neutralizing capability is indicated by neutralizing titer *, meaning the dilution factor needed to reduce antibody levels below detectable limits; IC_50_ ^†^, meaning half maximal inhibitory concentration; or NT_100_ ^‡^, meaning complete protection from cytotoxic effects of infection.

Antibody Name	Plant	Affected Lineages	Neutralizing Capability (Neutralizing Titer *, IC_50_ ^†^ or NT_100_ ^‡^)	Source
CR3022	*N. benthamiana*	Original strain	Fail to neutralize *	[71]
B38	*N. benthamiana*	Unidentified	640 at 0.492 µg/mL *	[90]
H4	*N. benthamiana*	Unidentified	40 at 5.45 µg/mL *	[90]
H4-IgG1-4	*N. benthamiana*	Unidentified	591 nM for H4-IgG3 ^‡^	[91]
CA1	*N. benthamiana*	Original strain, Delta	9.29 nM: Original ^†^ 89.87 nM: Delta ^†^	[87]
CB6	*N. benthamiana*	Original strain, Delta	0.93 nM: Original ^†^ 0.75 nM: Delta ^†^	[87]
11D7	*N. benthamiana*	Original strain, Delta, Omicron	25.37 µg/mL: Original ^†^ 59.52 µg/mL: Delta ^†^ 948.7 µg/mL: Omicron ^†^	[46]

**Table 3 life-13-00617-t003:** Compounds of medicinal plants found to be effective against SARS-CoV-2 through in vitro, in vivo or in silico analysis.

Major Active Compounds	Plant Species	Efficacy/Mechanism of Action *	References
Artemisinin, flavonoids, artesunate, artemether, nonidentified metabolites	*Artemisia annua* L.	Inhibiting viral replication	[159]
		Inhibiting replication of five virus variants including Delta	[160]
		Inhibiting viral infection	[161]
Astersaponin I (AI)	*Aster koraiensis*	Inhibiting virus entry pathways at plasma membrane and within endosomal compartments	[162]
Curcumin	*Curcuma longa*	Binding and inhibiting S protein of Omicron variant (in silico analysis)	[163]
Emetine	*Carapichea ipecacuanha*	Blocking viral entry into cells; inhibiting virus replication; anti-inflammation	[164]
Hesperidin (Hesperetin)	Various species	Reducing expression of TMPRSS2 and ACE2 (in silico analysis)	[165]
Hypericin	*Hypericum perforatum*	Binding viral envelope and reducing its infectivity	[166]
Licorice-saponin A3 (A3) and glycyrrhetinic acid (GA)	*Glycyrrhiza uralensis*	Inhibiting viral infection	[167]
Luteolin	Various species	Reducing viral replication by inhibition of RdRp	[168]
Myricetin	Various species	Inhibiting viral replication and transcription by inhibition of protease (Mpro); anti-inflammation (in vivo analysis)	[169]
Nonalkaloid compounds	*Rhazya stricta*	Binding key residues of S proteins and impeding viral infection (in silico analysis)	[170]
Panduratin A	*Boesenbergia rotunda*	Inhibiting viral replication	[171]
Persimmon-derived tannins	*Diospyros kaki*	Inhibiting virus replication; potential as a prophylactic agent	[172]
Piperine	*Piper* spp.	Inhibiting viral replication	[173]
Quercetin	Various species	Impeding viral replication by inhibition of RdRp	[168]
Thapsigargin	*Thapsia garganica*	Inhibiting viral replication by inducing stress in ER and increasing the viability of infected cells	[174]
Withaferin A, Withanone, Withanolide A	*Withania somnifera* (L.)	Inhibiting viral infection and replication; anti-inflammation and proinflammatory cytokines (in vivo analysis)	[175]

*: all those unindicated are in vitro analysis.

## Data Availability

Not applicable.

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
