# Peer review of "Plants as Biofactories for Therapeutic Proteins and Antiviral Compounds to Combat COVID-19"

_life, 2023, doi:10.3390/life13030617_

Round 1

Reviewer 1 Report

1.In page 1 and 2, line 36-45

The authors should provide proper scientific reference for the sentences of this part.

2.In page 2, line 47

You have mentioned the name of Figure 1 in this line but Figure 1 locates in page 4. Please add this figure in the nearest part after you mention its name for the first time.

3. In page 2 line 55

Plese reform multiple references in this section.

4. page 2, line 54 - 55

Please keep continuity between these two paragraphes. You should mention a sentence as a reletive link between one paragraph and its next one.

5.In page 2, line 56-59 and line 64-67

Why the authors did not consider suitable reference at the end of the sentence in this part?

6. In page 2 and 3, line 70- 98

This part of the manuscript is full of scientific findings with no scientific related references!

Please make sure that all sentences in this part have relevent references.

  7.Page 3, line 99-105

Why the sentences do not have references?

8. In page 3 line 123

Please reconsider the multiple references that are presented in this part.

9. In page 4 and 5, line 134-157, 

Some sentences have multiple references. On the contrary, some others have no any 

references. Please consider this note that every scientific facts, results and information in the text of the manuscript must have proper related reference.

10. In page 5, line 166-171

Proper references should be added.

11. In page 5, line 182

Multiple references should be reformed

12. In page 5, line 183

Table 1 should add at the place where you mrntion its name for the first time.

13. In page 5 and 6, line 186-194

This part needs references.

14. In page 6, line 199 and line 200

Multiple references need to be reformed.

15. In page 6, line 206

Why the authors have written this "(Rebelo et al. 2022)"?

16. In page 6, line 216

Multiple references should be reconsidered.

17. In page 6, line 211-218

What is the purpose of the authors by saying this part? And has this part of the manuscript disrupted the contiuity of the content of the part" 2.1.1. Plant-produced subunit vaccines "

Was it not better to rewrite this part so that it can keep consistency of the text?

18. In page 6, part "2.1.1. Plant-produced subunit vaccines "

Why the aithors have not mentioned any thing about conclusion of this part? Is it not more proper to conclude your information at the end of each section of the manuscript?

19. Page 6, line 222-227

The sentences need related referebce.

20. Page 7, line 243-246

Please add relevent reference.

 21. In page 7, line 253-265

Why the authors have mentioned this psrt of manuscript?

Please tell me the importance and necessity of this part and also the relationship of this part with the whole part of "2.1.2. Plant-produced VLP vaccines"

22. Why the authors have not written about the barriers or limittations for plants to be one of the most important sources of manufacturing anti-covid 19 vaccines?

23.why there is no comparison between plants and other sources of vaccines? The reason that other researchers have preferred 

to use other species rather than plants to produce vaccines?

24. In Table 1

In the part "Subunit"

The "Trade name" of three number of the cells of this table have is n/a. But why one of them has three references and the other two has one?

25. In page 8, line 290

Please insert Table 2, right after you mention its name for the first time.

26. Page 8 and 9 line 309-310 and line 315-326

Relevent references should be added at the end of each sentence.

27. Page 9, line 335-337 and line 341-342

Please add proper reference.

28. Page 9, line 345

Please reform multiple references

29. In page 10 line 351, 354, 380, 397 and 398

Multiple references should be reconsidered

30. Inoage 10 and 11, the part" 2.4. Plant produced antiviral lectins"

Why lectins derived from plants are not used for treatment of covide 19 in a vast scale yet?

31. In page 11, line 418-421

Please add proper reference at the end of each sentence

32. Page 11, line 434 and 436

Please reform multiple references

33. The part " 3.1. Antiviral mechanisms of PSMs" 

It would be better that the authors draw a Figure to summarize the this part and turn 

this part into a more comprehendable section

34. Why the authors have not mentioned the references of the sentences in part "3.1. Antiviral mechanisms of PSMs" in page 11, line 441-447?

35. Why authors have not mentioned animal studies in part "3.1. Antiviral mechanisms of PSMs", part "3.2. Major classes of PSMs against SARS-CoV-2 " and part" 3.3. Potential anti-SARS-CoV-2 compounds"

36. In page 14, line 556

Please insert Table 3 right after you mention its name fir the first time.

37. Page 15, line 607

Please reform middle-sentence reference and put each reference at its proper place (at the end of the sentences)

38. About Table 3

The colomn entitled "Efficacy/Mechanism of action" has long detailed sentences. Please reform this part of the table. Write brief (not long detailed sentences) in table but you should speak about this colomn in the main text of the manuscript in detail.

38. In page 17, line 645-646

What is the purpose of authors for writing this sentence "The COVID-19 pandemic has turned our own health and the world economy up-side down." ?

39. In page 17, line 644

Please separate the part " Prospects" from the part "conclusion" and write about each one separately (line 645-670 should be trasfered to the part "prospects")

40. Please talk about the "Prospects" in the use of plants in order to cure covid-19 and tell 

us about the future usages and problems of using plants and speak about future potential of plants in this field and also write about the commercial aspects of utilizing plant-derived compounds as drug for mentioned purpose.

41. In page 17, line 670-674

Please translocate this part and add it into the next paragraph

42. In the part "conclusion" authors should only speak about the facts that are resulted from their present findings. Thus, please rewrite the part "conclusions" and exert mentioned note

43. Why authors have inserted references in the part " conclusion"?

Please reconsider it.

In the part conclusion the authors must only speak about the facts that conclude from their present findings. (If they want to 

compare their findings with other studies, they should perform this comparison in the main text, not in the part "conclusion")

44. In page 18, line 483-486

The authors have mentioned some weak points of the use of plants in order to produce curative compounds.

My question is that why the authors have not spoken about it in the main text in the form of a separate heading, But they have written about it in the part "conclusion"?

45. In line 697-699 in page 18

The authors have mentioned some data about the toxicity of PSMs. This, why the authors have not written about the side effects of PSMs in the main text of the manuscript?

46. Why the authors have mentioned the line 

700-704 in page 18? 

47. All over the manuscript, there are some sentence without reference, some sentence with middle-sentence reference and some with multiple reference. 

The authors should reform all of them.

48. Please check reference list carefully (specially their titles)

Author Response

Dear Reviewer:

We appreciate all your invaluable comments/suggestions! The manuscript has been extensively revised to address all the concerns and suggestions. All the changes made in the manuscript are marked in red. A point-by-point response to specific comments is provided in the attached file. Your favorable consideration of this revised manuscript is appreciated.

Sincerely,

Jianfeng Xu

Reviewer 2 Report

I recommend this paper to be published in the journal. Here are some suggestions:

1: The paper is generally well written and structured. However, there are several review articles with the identical criteria and focus that are recently published. What motivated the authors to prepare a review. In “INTRODUCTION”, it is suggested to highlight the novelty of this work clearly.

2: Please add relevant literatures. “With the recent lifting of the "Zero-Covid Dynamics" policy in China, many more people will be infected, and mortality will continue to increase.(Nature Medicine, doi: 10.1038/s41591-023-02212-y)” “However, its immune evasiveness and high transmissibility pose a great threat to the global healthcare system (Front. Immunol. 2022, 13, 1015355).”; “Great efforts have been made in the past 2 to 3 years to counteract the spread of the virus through development of vaccines (Nature Immunol. 2022, 23, 360-370), immune-based therapy (Allergy. 2022, 77, 100-110), antiviral therapy (J. Med. Virol. 2022, 94, 1373-1390), and natural remedies (Pharmaceutics. 2021, 13, 1839; Biomedicines. 2021, 9, 689).”; “Emetine, an alkaloid extracted from the ipecacuanha plant (Brazilian root), is another compound with strong antiviral activity (Front. Pharmacol., 2020, 11, 1013)”; “Plants naturally produce a diverse range of bioactive small molecules, such as alkaloids (Molecules 2021, 26, 6171), flavonoids (J. Ethnopharmacol. 2021, 270, 113869), terpenoids (Biomedicines 2021, 9, 1505), and phenolic compounds (Int. J. Mol. Sci. 2022, 23, 2643), which are the source of countless pharmaceutical compounds for treating various diseases.”

3: “Conclusions” of the manuscript. To improve the use of natural products, combination treatment that exhibit better additive or synergistic effects against COVID-19 is a promising strategy. “Plant natural products have demonstrated potential value and with the assistance of nanotechnology and combination drug therapies, this natural remedy could serve as a starting point for further drug development in treating this diseases.” An additional description of relevant content would be beneficial.

4: Chemical structures of bioactive natural products should be added.

Author Response

Dear Reviewer:

We appreciate your invaluable comments/suggestions! The manuscript has been extensively revised to address all the concerns and suggestions. All the changes made in the manuscript are marked in red. A point-by-point response to specific comments is provided in the attached file. Your favorable consideration of this revised manuscript is appreciated.

Sincerely,

Jianfeng Xu

Round 2

Reviewer 1 Report

It is now acceptable for publication.